# Real-time observation of a correlation-driven sub 3 fs charge migration in ionised adenine

Erik P. Månsson [1,2], Simone Latini [3], Fabio Covito[3], Vincent Wanie [1,2,4], Mara Galli[1,5], Enrico Perfetto[6,7], Gianluca Stefanucci[7,8], Hannes Hübener [3], Umberto De Giovannini [3,9], Mattea C. Castrovilli[2,10], Andrea Trabattoni[1], Fabio Frassetto[11], Luca Poletto[11], Jason B. Greenwood[12], François Légaré [4], Mauro Nisoli [2,5], Angel Rubio [3,13] & Francesca Calegari [1,2,14 ✉]

Sudden ionisation of a relatively large molecule can initiate a correlation-driven process dubbed charge migration, where the electron density distribution is expected to rapidly move along the molecular backbone. Capturing this few-femtosecond or attosecond charge redistribution would represent the real-time observation of electron correlation in a molecule with the enticing prospect of following the energy flow from a single excited electron to the other coupled electrons in the system. Here, we report a time-resolved study of the correlation-driven charge migration process occurring in the nucleic-acid base adenine after ionisation with a 15–35 eV attosecond pulse. We find that the production of intact doubly charged adenine – via a shortly-delayed laser-induced second ionisation event – represents the signature of a charge inflation mechanism resulting from many-body excitation. This conclusion is supported by first-principles time-dependent simulations. These findings may contribute to the control of molecular reactivity at the electronic, few-femtosecond time scale.

[1] Center for Free-Electron Laser Science CFEL, Deutsches Elektronen-Synchrotron DESY, Hamburg, Germany. [2] Institute for Photonics and Nanotechnologies CNR-IFN, Milano, Italy. [3] Max Planck Institute for the Structure and Dynamics of Matter and Center for Free Electron Laser Science, Hamburg, Germany. [4] INRS-EMT, Varennes, QC, Canada. [5] Department of Physics, Politecnico di Milano, Milano, Italy. [6] CNR-ISM, Division of Ultrafast Processes in Materials (FLASHit), Monterotondo Scalo, Italy. [7] Dipartimento di Fisica, Università di Roma Tor Vergata, Roma, Italy. [8] INFN, Sezione di Roma Tor Vergata, Roma, Italy. [9] Dipartimento di Fisica e Chimica, Università degli Studi di Palermo, Palermo, Italy. [10] Institute for the Structure of Matter CNR-ISM, Monterotondo Scalo, Italy. [11] Institute for Photonics and Nanotechnologies CNR-IFN, Padova, Italy. [12] Centre for Plasma Physics, School of Maths and Physics, Queen's University Belfast, Belfast, UK. [13] Center for Computational Quantum Physics (CCQ), The Flatiron Institute, New York, NY, USA. [14] Institut fur Experimentalphysik, Universität Hamburg, Hamburg, Germany. ✉email: francesca.calegari@desy.de

The interaction of ionising radiation with molecules often leads to an internal electronic rearrangement, governed by correlated processes such as shake-up or Auger[1,2] and Interatomic Coulombic Decay (ICD)[3]. The superposition of electronic states resulting from the many-body excitation has been predicted to initiate attosecond charge migration along the molecular backbone, when the nuclei can be still considered as frozen[4–7]. In the last few decades, advances in the development of extreme ultraviolet (XUV) attosecond light sources[8,9] have given access to electron migration in molecules, holding a great promise for attochemistry. In this context, a signature of few-femtosecond charge dynamics has been identified in aromatic amino acids by exploiting the characteristic broadband of the attosecond radiation to create a coherent superposition of cationic eigenstates[10,11]. Nevertheless, the originally conceived charge migration process[4], depicted as a non-stationary charge distribution resulting from the removal of an electron from a correlated state, remains to be demonstrated. This would constitute a unique route for mapping in real-time the energy flowing from the single excited electron to all other coupled electrons in the molecule, i.e., the intrinsic nature of the electron correlation and the resulting dynamics[12]. The above-described purely electronic scenario would only survive until the nuclei start to move, i.e., typically <10 fs from the ionisation event[13–15]. Therefore, one would need to act on the system very promptly after the ionisation event to take advantage of this fast charge redistribution and achieve control over the molecular reactivity. On a time scale of several tens of femtoseconds multi-electronic and non-adiabatic effects are fully entangled and their interplay has been recently identified in the relaxation dynamics following XUV-induced ionisation of organic molecules[16,17].

Our few-femtosecond time-resolved study has the motivation of tracking many-body effects in real-time before non-adiabatic effects take place. In this work, we report—to the best of our knowledge—on the first experimental evidence of correlation-driven charge migration, occurring in the nucleic-acid base adenine after sudden ionisation by an XUV attosecond pulse, and we show how to take advantage of electronic correlations to obtain an ultrafast control "knob" for the molecular dissociation.

## Results and discussion

**Time-resolved experiment**. In our experiment, ionisation of adenine is initiated by an isolated sub-300 as XUV pulse containing photon energies from 15 to 35 eV, produced by high-harmonic generation[18] in krypton gas through the polarisation gating technique[19] (a typical XUV spectrum is shown in Supplementary Fig. S1). A waveform-controlled 4-fs near-infrared (NIR, central photon energy 1.77 eV) probing pulse is combined with the XUV pump pulse using an interferometric approach. Adenine is sublimated and carried to the laser interaction region by using helium as a buffer gas. The produced ions are then collected as a function of the XUV-pump NIR-probe delay (see Fig. 1a), using a time-of-flight spectrometer. The ion mass spectrum resulting from ionisation by the XUV pulse is dominated by ionic fragments (81% of the total yield as calculated in Supplementary Methods, section 2 and Supplementary Fig. S2), indicating a relatively low photostability of the molecule in this energy range. Further deposition of energy by the NIR pulse, simultaneously or after the XUV, leads to an overall increase of fragmentation[20]. Figure 1b shows the partial ion yield for several ionic fragments as a function of the pump–probe delay. The time-dependent yields of the cationic fragments with mass 27, 38 and 53 u display step-like increases, followed by slower decay. The enhancement of small fragment ions occurs at the expense of the

large fragments, mainly 108 u, which clearly indicates that the combination of XUV and NIR pulses leads to further excitation and therefore more efficient bond breaking.

The most intriguing observation in the time-dependent mass spectrum is the appearance of a new ion for small positive delays at mass/charge = 67.5 u/e, corresponding to the doubly charged parent molecule (adenine$^{2+}$)[21–23]. It is worth noting that a stable dication of the parent is difficult to discern in the XUV-only signal or at negative NIR delays, and none is present if we select the portion of the XUV spectrum below 17 eV (see Supplementary Methods, section 3 and Supplementary Figs. S3 and S4). Figure 1b shows that the formation of the parent dication is delayed compared to the cationic fragments. Fitting the experimental data using a curve model described in Supplementary Methods, section 5, we obtain for the dication pump–probe signal a delay of $2.32 \pm 0.45$ fs ($\tau_1$, exponential risetime) and a decay time of $24 \pm 3$ fs ($\tau_2$, exponential decay) (for more details see Supplementary Fig. S6). To further verify that the steps of the cationic fragments accurately represent the absolute zero time delay (XUV–NIR overlap), we also did measurements with simultaneous injection of an atomic gas (krypton). The XUV + NIR double ionisation of krypton gives a Kr$^{2+}$ signal at a time consistent with the adenine cations (see Supplementary Methods, section 6 and Supplementary Fig. S7), allowing us to conclude that it is the adenine dication signal, which is positively delayed. The extracted delay does not appear to depend on the NIR pulse intensity, within the explored range from $7 \times 10^{12}$ to $1.4 \times 10^{13}$ W/cm$^2$. At the same time we observed that the dication yield scales quadratically with the NIR intensity (see Supplementary Methods, section 4 and Supplementary Fig. S5), thus indicating that two NIR photons are required to doubly ionise the molecule.

**Interpretation of the dynamics**. The detection of a doubly charged ion with a sub-3 fs delay is a probe for a pure electronic mechanism initiated by the XUV sudden ionisation and occurring before the control NIR laser pulse arrives. It is worth mentioning that, the possibility of combined electronic and nuclear dynamics (non-adiabatic effects such as conical intersections[24]) cannot be ruled out a priori. Nevertheless, in the sub-3 fs time window, we do not expect these effects to be significant. This conclusion is supported by first-principles calculations[25–27]—based on time-dependent density functional theory (TDDFT)[28,29] and Ehrenfest dynamics[30]— indicating that the nuclei can be almost considered as frozen in this short time scale. As an example, in Fig. 1c we report the time evolution of the bond lengths after removal of an electron from the fourth highest occupied molecular orbital (a similar analysis where the electron is removed from different occupied orbitals is reported in Supplementary Methods, section 8 and Fig. S10 and leads to the same conclusions). Several bond lengths are seen to evolve and drift away from their equilibrium value only after 3 fs.

Having in mind that the correlated electron dynamics may have a primary role in the delayed creation of the dication, we propose the following mechanism: (I) the XUV pulse singly ionises the molecule leaving a hole in an inner valence state, (II) the hole decays in a characteristic transition time and, due to electronic correlations, this can lead to excitation of a second electron to a bound excited state, hereafter called shake-up state, (III) the NIR pulse extracts the excited electron, hence doubly ionising the molecule. A simplified representation of this scenario is pictorially illustrated in Fig. 2. As anticipated, the removal of an electron from a correlated state may result in a non-stationary charge distribution that rapidly evolves in

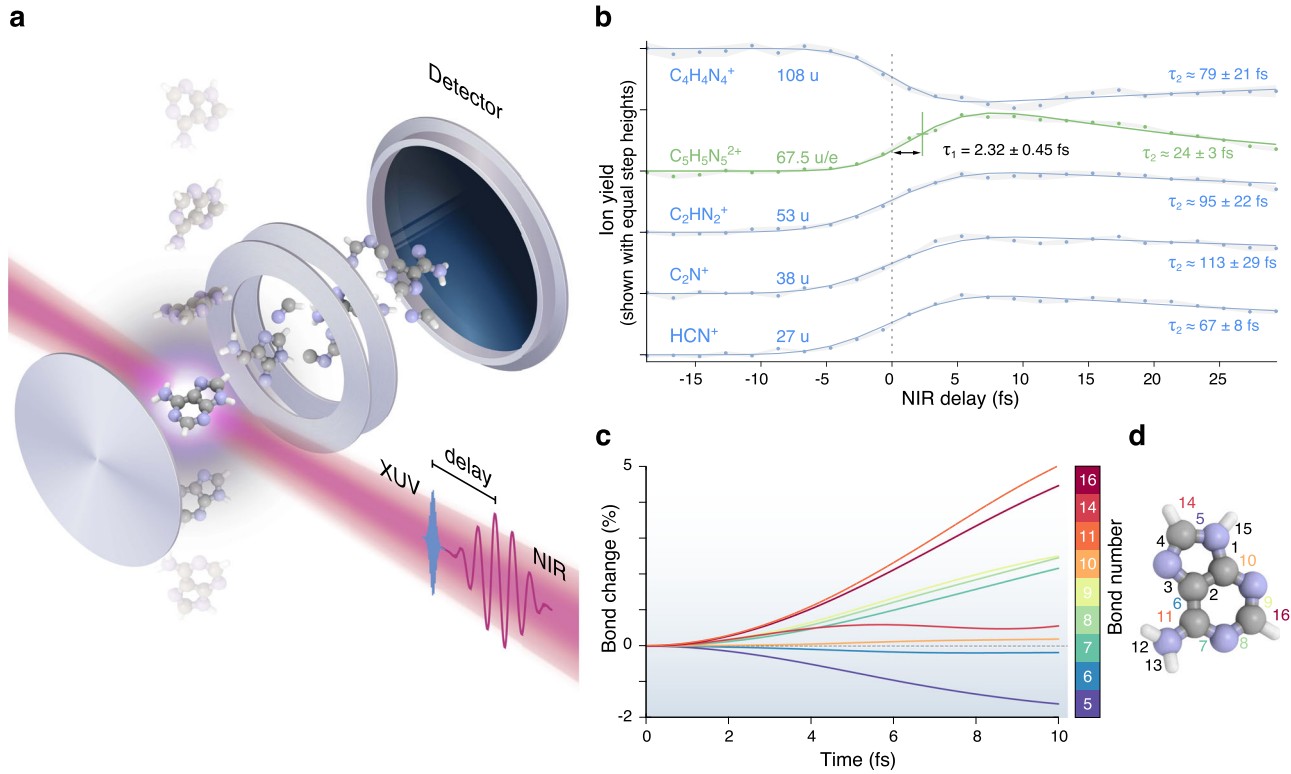

**Fig. 1 Experimental results: pump–probe scan. a** Schematic of the experiment: a molecular beam is injected into a VMI operated in the ion time-of-flight mode. Adenine ions are accelerated towards the detector and the ion yield is measured as a function of the XUV-pump IR-probe delay. **b** Normalised yield of several ions (shown with a vertical offset) as a function of the XUV-pump NIR-probe delay. The yield of the ionic fragments exhibits a distinct positive or negative step-like behaviour, while the adenine parent dication (67.5 u/e) is fitted to have an exponential risetime of $\tau_1 = 2.32 \pm 0.45$ fs (68 % confidence interval). The decay lifetime ($\tau_2$) is significantly shorter for the dication (green curve) than for the cations. The grey shading indicates the standard error of the mean of seven successive scans. **c** Example of calculated time evolution of bond lengths in the first 10 fs following XUV ionisation, with an electron removed from the fourth highest occupied molecular orbital (HOMO − 3). All the bonds start elongating only after 3 fs. The theoretical simulations for bond elongation are performed with TDDFT. **d** The bond numbering used in the theoretical work (for more details see Supplementary Methods, section 7, Supplementary Figs. S8 and S9).

time (charge migration). We could already speculate that the initiated charge migration process determines the optimal time window for an increased absorption of the NIR probe pulse. Moreover, the creation of the dication can only take place after the shake-up process has occurred but before the excited molecular cationic state relaxes via non-adiabatic couplings. Therefore, the delay-dependent dication signal works as a precise clock for the above-described correlation-driven process.

To corroborate our interpretation, we first evaluated characteristic shake-up times and searched for a peculiar one compatible with the experimentally observed time delay. The shake-up process illustrated in the level diagrams of Fig. 2a is purely driven by electronic correlation (two-body Coulomb interaction), not accounted for in standard TDDFT simulations using adiabatic exchange correlation functionals[31]. Nevertheless, a simple estimation of the shake-up transition time can be obtained with a rate equation approach: the initial statistical superposition of states created by the XUV pulse is calculated using ab-initio photo-ionisation probabilities, and Fermi's golden rule is used to obtain the shake-up rate due to the Coulomb interaction (see Supplementary Methods, section 9 and Figs. S11–S13). Figure 2b shows the characteristic shake-up transition times towards different bound excited orbitals (Kohn–Sham (KS) orbitals obtained with density functional theory ground-state calculations). Most of the values are of the order of a few hundreds attoseconds except

for three states, one of which (the LUMO+6 indicated in green) is 2.5 fs, very close to the experimentally observed delay of the parent dication formation. Interestingly, the energy of this orbital is in the window of two-photon ionisation from the NIR pulse, which is in agreement with the above-mentioned experimental observation that two NIR photons are required for the creation of the dication.

**Many-body time-dependent simulations**. While the rate-equation approach is intuitive and it provides a clear physical explanation of the experimental findings, it is an overly simplified description since it treats electronic correlations in first order perturbation theory, in a non-dynamical fashion and lacks a first principles description of the XUV ionisation process. More refined and independent ab-initio calculations are required to further validate our interpretation and provide a predictive framework to address similar phenomena in other molecules. To this end, we performed many-body time-dependent simulations from first-principles to take into account both the electron dynamics triggered by the XUV photoionisation and the absorption of a delayed NIR pulse. By solving the equations of motion for the non-equilibrium Green's function and using an efficient propagation scheme based on the generalised Kadanoff–Baym ansatz, we can obtain an accurate and controlled treatment of shake-up processes[32,33] and describe the light-molecule interaction from first-principles using laser

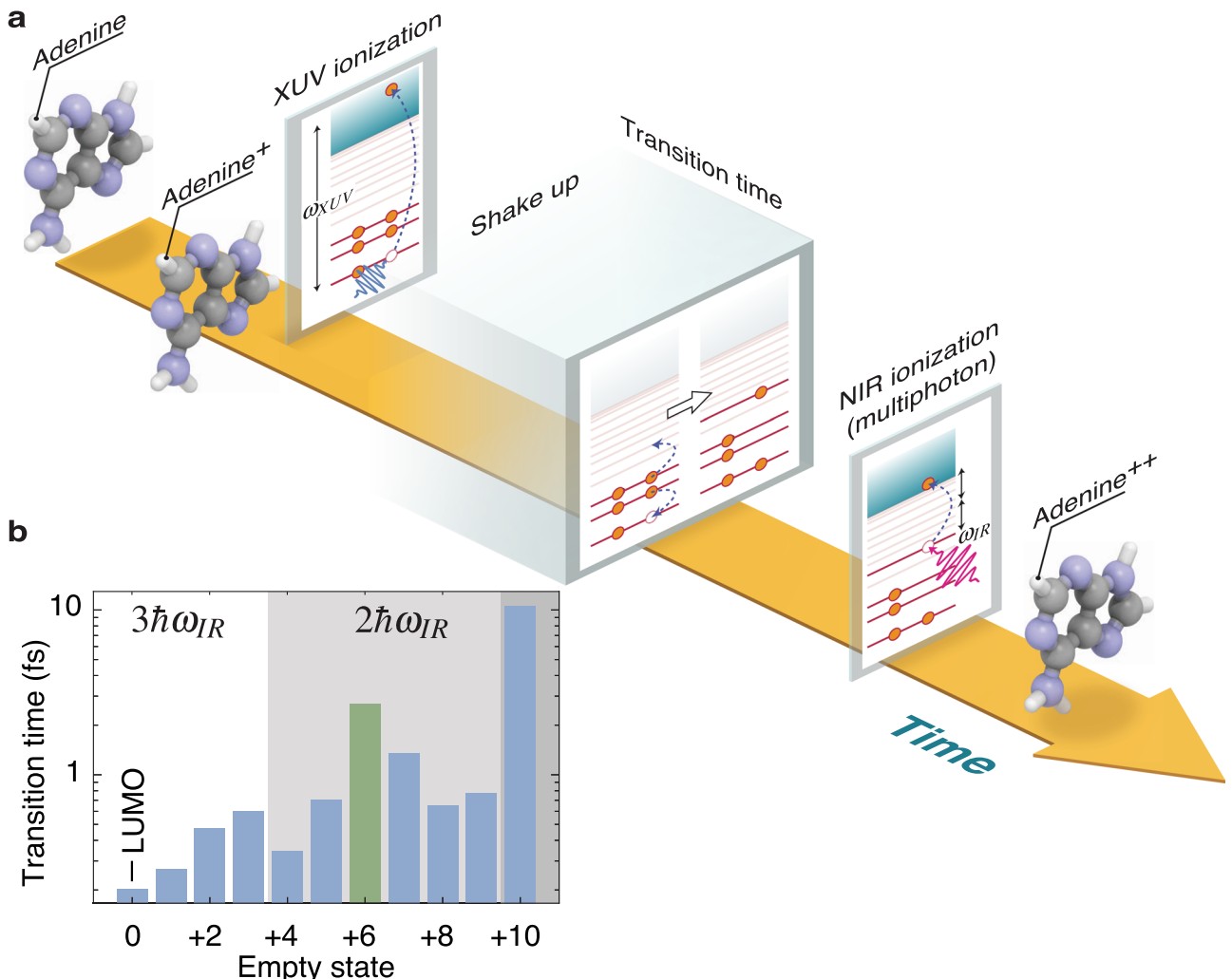

**Fig. 2 Overview of the molecular dynamics: the shake-up process. a** Following XUV photoionisation, a hole is created in the inner valence. The hole decays in a characteristic transition time and, due to electronic correlations, this can lead to excitation of a second electron to a bound excited state, called shake-up state. If optimally time-delayed from the XUV, the NIR control pulse extracts the excited electron, hence doubly ionising the molecule. **b** Transition times to a given shake-up state calculated with a Fermi's Golden rule approach. A special shake-up state (LUMO+6) is highlighted in green and shows a characteristic time of 2.5 fs. The states are ordered by energy and grouped (shades of grey) by the number of NIR photons (1, 2 or 3) required to ionise a second electron. The LUMO+6 state falls in the two NIR-photon group.

pulses with the same characteristics as in the experiment (see Supplementary Methods, sections 10 and 11 and Figs. S14–S18). It is important to point out that our simulations are well suited to describe the electron dynamics but do not take into account the nuclear dynamics and therefore non-adiabatic effects (e.g., conical intersections). From the calculations, we first extract the orbital-resolved occupations that are reported in Fig. 3a. When only the XUV photoionisation is considered, we can confirm the results obtained with the rate equations: while most of the states exhibit sub-femtosecond rise-times, the LUMO+6 occupation (shown in green) rises over several femtoseconds due to a slow shake-up process. Figure 3 shows the integrated time-dependent electron density more than 3 Å away from the molecular plane (panel (b)) and snapshots of the change in electron density (panel (c)). As it can be observed from this figure, a significant electronic charge inflation builds up over a few-femtoseconds. The spatial distribution of this density variation resembles the one of the LUMO+6 orbital (Supplementary Fig. S17), high-lighting its dominant role in the overall electron dynamics. From Fig. 3b it could be seen that the integrated electron density

rapidly increases in the first 3 fs and therefore we could argue that this rapid delocalisation far from the molecular plane plays a dominant role in the delayed absorbtion of the NIR probe pulse for the efficient creation of the stable dication. Our simulations also indicate that LUMO+6 can only be accessed when ionisation is triggered by an XUV pulse polarised per-pendicularly to the molecular plane. Therefore, the relative orientation between the molecule and the attosecond pulse can potentially be exploited to achieve more refined control over the ionisation process.

Finally, we have included the NIR absorption in the simulation and calculated the time-resolved NIR-induced depletion, i.e., population reduction, of the LUMO+6 (Fig. 4a), for different pump–probe delays. The depletion shows an onset in the window of 2–4 fs and increases with larger delays. To clarify the delay dependence of the depletion, we present the average depletion (over a 1 fs window following the NIR pulse) as a function of pump–probe delay in Fig. 4b. The trend reproduces, remarkably, the one measured for the adenine dication yield (green solid line in Fig. 2). We point out that only the LUMO+6 state is

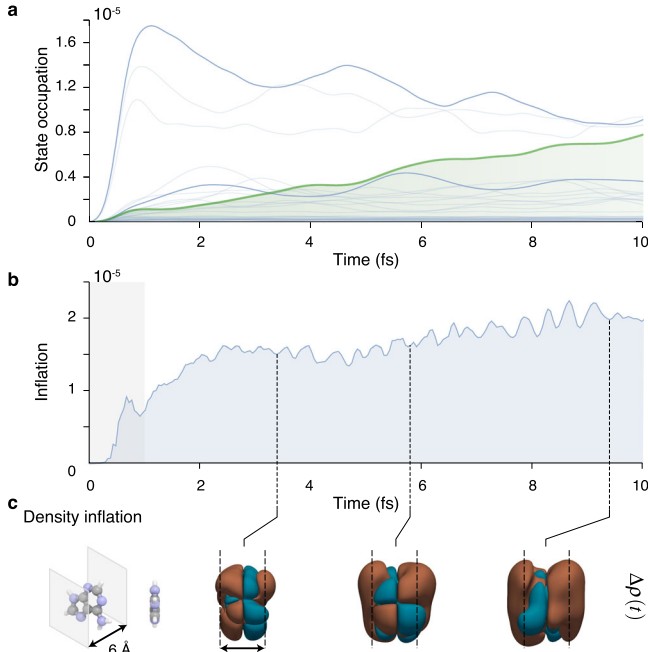

**Fig. 3 Theoretical results: shake-up and charge inflation. a** Time-dependent occupations of the adenine bound excited states after photoionisation by the XUV pulse, calculated with the ab-initio non-equilibrium Green's function method. The special state (LUMO+6), highlighted in green, is populated via the shake-up process and its population rises over several femtoseconds to one of the largest values. **b** Integrated time-dependent electron density more than 3 Å away from the molecular plane. The grey shaded area represents the time-window of the pump pulse, having its peak at $t_{pump} = 0.48$ fs. **c** Left panel: the adenine molecule and the planes defining the integration region. Right panel: snapshots of the variation of the electronic density with respect to the density immediately after the XUV pulse. We observe that the electronic density inflates considerably away from the molecular plane. The noticeable out-of-plane charge migration can be attributed to the increasing population of the correlated LUMO+6 state (Supplementary Fig. S17). These results have been obtained with the non-equilibrium Green's function method.

characterised by this slow onset (other states in Supplementary Fig. S18) and we can therefore conclude that this peculiar shake-up dynamics—resulting in the out-of-plane charge migration mechanism—explains why the NIR pulse has to be optimally delayed in order to produce the stable dication. We note that our model cannot reproduce the 24-fs exponential decay observed in the time-dependent dication yield, since non-adiabatic couplings have been neglected. We presume, however, that this decay is a signature of the dephasing induced by the recently depicted electron-phonon like coupling occurring in correlation bands for large molecules[17].

## Conclusions
To summarise, our theoretical calculations singled out a special shake-up state (LUMO+6) leading to an out-of-plane charge migration mechanism, which mediates the observed delayed creation of stable doubly charged adenine (remaining intact for at least the time it takes to be detected by our mass spectrometer, i.e., a few microseconds). The peculiarity of this state can be attributed to the following characteristics: (I) it has a few-femtoseconds shake-up time, compatible with the experimentally observed delay in the dication formation, (II) it is a delocalised excited state that extends away from the molecular

plane and (III) it couples very efficiently to the NIR pulse. Our findings not only indicate that the delayed creation of the dication is a valuable probe for a many-body effect, but also that by precisely timing a NIR control pulse one could take advantage of a correlation-driven charge redistribution to prevent dissociative relaxation.

Real-time mapping of the many-body effects in ionised biochemically relevant molecules is the first (crucial) step towards a control of the molecular photo-reactivity at the electronic time scale. Here, we have not only characterised the intrinsic time for a shake-up process to occur in ionised adenine but also the resulting out-of-plane charge migration, neither of which to our knowledge has been measured in real-time for any polyatomic molecule before. A key aspect here is that the many-body effect mediate the efficient absorption of a properly delayed NIR control pulse, leading to the creation of intact and stable doubly charged adenine. Furthermore, our findings demonstrate that extreme time resolution is required to act promptly after molecular ionisation and take advantage of the non-equilibrium charge redistribution (before non-adiabatic effects take place) to achieve control over the molecular dissociation. By complementing the experiments with covariant detection of electrons and ions and with the support of more advanced time-dependent many-body methods, including the nuclear motion, we could potentially obtain a direct mapping of this electronic redistribution and target more specifically new photoreaction pathways for a wide range of polyatomic molecules.

## Methods
Carrier-envelope-phase stable laser pulses with 4 fs duration and a central wavelength of 700 nm are used to drive the attosecond pump–probe setup[10]. Seventy percent of the 2.5 mJ energy is used for attosecond pulse generation in krypton using the polarisation gating technique[19]. The remaining part of the NIR pulse is recombined after an adjustable delay stage and used as probe. Further details are given in Supplementary Methods, section 1. The adenine powder sample (Sigma-Aldrich, > 99 %) is evaporated at 463 K in a resistively heated stainless steel oven, with a flow of helium acting as carrier and buffer cooling gas. Cations are detected in a time of flight mass spectrometer where the detector is gated to block the helium signal at short times of flight.

Pump–probe data are acquired by scanning the delay in alternating directions over multiple traces. The average of the traces is fitted with a model based on rate-equations, either as the population of a state directly populated by the XUV pulse and exponentially decaying with the lifetime $\tau_1$, or as a system of opposing rates where the probed population starts at zero and rises with lifetime $\tau_1$ and decays with a slower lifetime $\tau_2$. More details are given in Supplementary Methods, section 5. A sequence of short scans at different NIR intensities was used to estimate the number of NIR photons involved in the processes. A scan with simultaneous injection of krypton was also performed to confirm that the zero time delay is where the pump–probe scan shows enhancement of adenine cationic fragments.

The stability of the cation and dication are confirmed by ab-initio calculations when geometry relaxation, i.e., energy dissipation, is allowed. The ab-initio calculations are performed using Octopus[27] within a DFT framework, using the PBE functional together with the averaged density self-interaction correction (ADSIC). The bond elongation following vertical ionisation by the XUV pulse is estimated via TDDFT + Ehrenfest dynamics after suddenly removing one electron.

A first estimate of the shake-up rates is obtained by applying Fermi's golden rule. The initial and final shake-up states involve many-body wavefunctions that differ by two single-particle states calculated with DFT using the PBE functional and the ADSIC. The Coulomb interaction is taken as perturbation and all the possible shake-up combinations, leading to the occupation of a given excited bound state via shake-up are statistically summed up. The inverse of the shake-up rate is then defined as the characteristic shake-up time. More details are provided in Supplementary Methods, section 9.

The more accurate full electronic dynamics simulation is based on the non-equilibrium Green's function formalism (more details in Supplementary Methods, section 10) and solves the Kadanoff–Baym equations numerically[32,33] within the Generalised Kadanoff–Baym Ansatz at the second Born level. Kohn–Sham orbitals up to 41 eV are used in a 20 Bohr sphere with a 0.25 Bohr grid spacing. From these orbitals a Hartree-Fock basis is generated and employed for the time propagation (more details in Supplementary Methods, section 11). The experimental XUV and NIR laser shapes are used in the simulations, which guarantees and hence the laser-induced coherence is included in our simulations. Given the moderate XUV energy range, we neglect Auger decay processes.

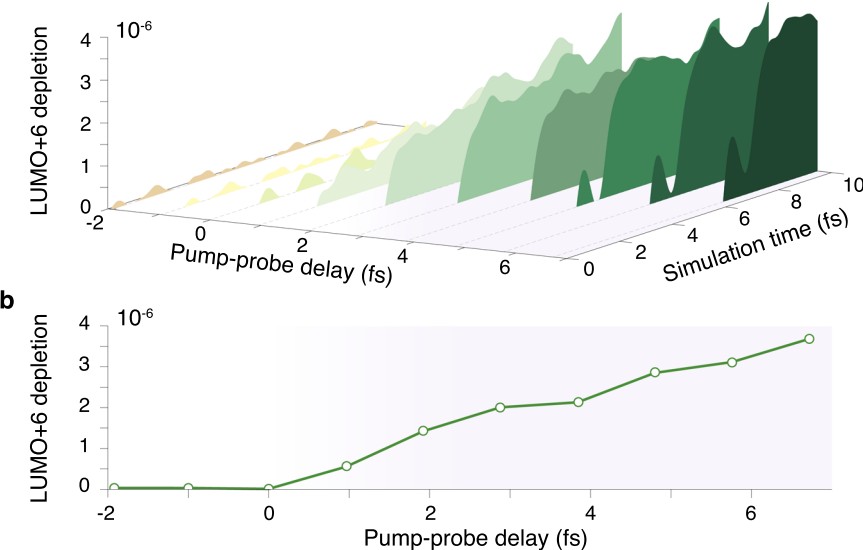

**Fig. 4 Role of the NIR probe pulse. a** Temporal evolution of the LUMO+6 state depletion due to the combined action of XUV and NIR pulses as a function of the delay. **b** Simplified readout with the state depletion averaged in a 1 fs window after the NIR pulse. The signal is absent at zero delay although half the NIR pulse then exposes the sample after the centre of the XUV pulse, it then shows a significant onset in the window of 2–4 fs and keeps increasing with larger delays. The trend reported in **b** reproduces, remarkably, the one of the time-dependent yield measured for the adenine dication. These results have been obtained with the non-equilibrium Green's function method.

## Data availability
The data that support the findings of this study are available from the corresponding author upon reasonable request.

## Code availability
The programme code used to treat experimental data is available from the corresponding author upon reasonable request. The simulations of electronic dynamics were done with previously described packages[32].

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

## Acknowledgements

F. Calegari acknowledges support from the European Research Council under the ERC-2014-StG STARLIGHT (Grant Agreement No. 637756). F. Calegari and A.R. acknowledge support from the Deutsche Forschungsgemeinschaft (DFG, German Research Foundation)—SFB-925—project 170620586 and the CUI: Advanced Imaging of Matter' of the Deutsche Forschungsgemeinschaft (DFG) - EXC 2056 - project ID 390715994. F.L. and V.W. acknowledge the Fonds de recherche du Québec—Nature et technologies (FRQNT) and the National Science and Engineering Research Council (NSERC). V.W. acknowledges support from the Vanier Canada Graduate Scholarship (Vanier CGS) programme. S.L. acknowledges support from the Alexander von Humboldt foundation. A.R. acknowledge financial support from the European Research Council(ERC-2015-AdG-694097). The Flatiron Institute is a division of the Simons Foundation. G.S. and E.P. acknowledge EC funding through the RISE Co-ExAN (Grant No. GA644076), the European Union project MaX Materials design at the eXascale H2020- EINFRA-2015-1, Grant Agreement No. 676598, Nanoscience Foundries and Fine Analysis-Europe H2020-INFRAIA-2014-2015, Grant Agreement No. 654360 and Tor Vergata University for financial support through the Mission Sustainability Project 2DUTOPI. J.B.G. acknowledge support from the EPSRC (UK) grant number EP/M001644/1. The authors would like to acknowledge A. Stolow for the fruitful discussion.

## Author contributions

S.L., E.P, G.S., L.P., F.L., M.N., A.R. and F. Calegari supervised the project. E.M., V.W., M.G. and M.C. performed the measurements, with F.F. and J.G. contributing resources. S.L., F. Covito, E.P., G.S. and H.H. performed the simulations. E.M., S.L., F. Covito, U.D., A.T., A.R. and F. Calegari wrote the manuscript. All authors contributed to the discussions and in improving the writing of the manuscript.

## Funding

## Competing interests

The authors declare no competing interests.
