## [Peer Review File · Communications Chemistry]

This manuscript has been previously reviewed at another Nature Research journal. This document only contains reviewer comments and rebuttal letters for versions considered at Communications Chemistry.

REVIEWERS' COMMENTS:

Reviewer #4 (Remarks to the Author):

The authors have addressed two main criticisms with their new version of the manuscript:

- 1) Limited significance of their work for photo stability of DNA.
- 2) Scepticism as to whether the reported mechanism (shake-up after ionization and removal of a loosely bound electron by a consecutive IR pulse) is general enough to motivate developing strategies for controlling chemical reactions.

The authors, in my view, have adequately addressed these points and the revised version of their manuscript now reflects a good balance of new insight and potential high impact in the AMO and chemistry fields.

Specifically, regarding aspects 1) and 2):

1) The authors weakened or removed arguments relating to the significance of their work for photo-protection/stability of DNA. I agree with the criticisms raised that such as gas-phase study of this particular ionization effect is limited in these terms. Most relation to DNA photo-stability was removed by the authors as indicated in their rebuttal. I also agree with the authors that their findings on electronic correlation measured in real time and potential of this for controlling chemical reactions should be of great interest to a broad community well beyond the AOM community. I therefore strongly support their claim that this manuscript is well suited for for the bread readership of Communications Chemistry.

2) Shake-up processes are ubiquitous and I see no reason for why the reported mechanism should not be the foundation for a general mechanism to control reactions. One certainly cannot generalize from one to many or all systems and that is not what the authors do. They report in a very clear concise way a new mechanism for how electron correlation may be used to steer chemical reactions and that per se is very nice, of broad interest to well beyond the AOM community and hence deserves publication in a journal with a broad readership. Whether the reported effect will prove to be general in nature will have to be seen, let us just see whether and how this evolves.

I think that this is a very nice study, written in a very clear way and with well substantiated claims. I believe that it will find appeal in a broad readership and I am looking forward to follow-up studies, both experimental and theoretical, by the same and other teams. I can therefore without any doubt recommend publication of the manuscript as is in Communications Chemistry.

Reviewer #5 (Remarks to the Author):

I think the authors properly addressed the questions raised by the previous Reviewers. Therefore I think the manuscript can be accepted for publication, and personally I think is a nice piece of work (although it does not seem to me these results are really relevant to DNA in real world so I guess previous referees were right asking to modify the focus and the presentation of the results)

According to what was requested to me by the Editor, I mostly limit to the above opinion.

I add just one comment: Do the authors excite a coherent electronic wavepacket or not? I guess so. In this case, is it reasonable to assume that in few femtoseconds it becomes a statistical mixture (this is the assumption behind the usage of Fermi Golden rule, although the adopted by the authors "statistical superposition" sounds a bit unclear to me).

Does the Green function treatment properly address the time evolution of coherences? Few comments in the SI could be added

I spotted few things the authors may want to improve in the SI:

Figure S9 the y axis needs units (Angstrom?)

Eq 14 I assume η is γ ...Which is the value they use for the Lorentzian broadening, how was it selected?

Page 9, below Eq 21. Hartree and not Hatree

Response to Reviewer #5

I think the authors properly addressed the questions raised by the previous Reviewers. Therefore I think the manuscript can be accepted for publication, and personally I think is a nice piece of work (although it does not seem to me these results are really relevant to DNA in real world so I guess previous referees were right asking to modify the focus and the presentation of the results)

We thank the Referee for acknowledging the importance of our work.

According to what was requested to me by the Editor, I mostly limit to the above opinion.

I add just one comment: Do the authors excite a coherent electronic wavepacket or not? I guess so. In this case, is it reasonable to assume that in few femtoseconds it becomes a statistical mixture (this is the assumption behind the usage of Fermi Golden rule, although the adopted by the authors "statistical superposition" sounds a bit unclear to me).

Does the Green function treatment properly address the time evolution of coherences? Few comments in the SI could be added

Our approach - based on non-equilibrium Green's function - allows us to fully include the electronic coherence induced by the laser excitation. A realistic, experimental-like, laser shape is used to excite the adenine molecule and induce a coherent superposition of electronic states. While in the present work coherence does not result in charge migration, other works from some of the authors [e.g., Science, 346, 336–339 (2014)] have shown that in other molecular systems, where correlations play a minor role, the electronic coherence induced by the laser is of key importance. For those systems, it has been demonstrated that the theoretical approach based on non-equilibrium Green's function can perfectly reproduce the dynamics induced by the electronic coherence [Journal of Physical Chemistry Letters 9, 1353 (2018), Journal of Physical Chemistry Letters 11, 891 (2020)]. We stress that in the Fermi Golden rule approach no laser induced coherence is included and the initial ionised state is a mere statistical mixture of singly ionised consistent with the energy window spanned by the XUV laser.

We have added the following sentence in the SI:

"By exciting the molecule with a realistic pulse allows us to take into account the laser induced electronic coherence. In refs. [Journal of Physical Chemistry Letters 9, 1353 (2018), Journal of Physical Chemistry Letters 11, 891 (2020)], where an equivalent theoretical approach is used, the effect of laser induced coherence is shown to affect the evolution of charge migration."

I spotted few things the authors may want to improve in the SI:
Figure S9 the y axis needs units (Angstrom?)

We thank the Referee for finding this mistake. We have corrected the axis label to include [a.u.] for atomic units (0.53 Å).

Eq 14 I assume η is γ ...Which is the value they use for the Lorentzian broadening, how was it selected?

The Referee's assumption is correct. We choose this particular value of η as the smallest broadening such that the results can still be considered independent on such choice. In this way we avoid the inclusion of unrealistic shake-up processes due to a too loose energy matching criterium.

To clarify this point we have added the following sentence in the SI:

“In FigS13(a) we show the characteristic electronic shake-up times as reported in the main text. For more thorough validation of this result we computed the same quantities as a function of the artificial parameter η . These results are reported in Fig S13(b) for the final empty states ionised with two IR photons and it is apparent that the variation of the characteristic times with respect to η is small for any $\eta \geq 0.1$ eV. In our calculations we chose $\eta = 0.1$, which is safe to assume considering typical quasiparticle lifetimes in molecules \cite{Caruso2013}. Furthermore, we choose this particular value as the smallest broadening such that the results can still be considered independent on such choice. In this way we avoid the inclusion of unrealistic shake-up processes due to a too loose energy matching criterium.”

Page 9, below Eq 21. Hartree and not Hatree

We thank the Referee for finding this typo, that we have now corrected.